



# Quality Aspects of the Wegener Center Multi-Satellite GPS Radio Occultation Record OPSv5.6

Barbara Angerer[1,2], Florian Ladstädter[1,2], Barbara Scherllin-Pirscher[3], Marc Schwärz[1,2], Andrea K. Steiner[1,2], Ulrich Foelsche[2,1], and Gottfried Kirchengast[1,2]

[1]Wegener Center for Climate and Global Change (WEGC), University of Graz, Graz, Austria
[2]Institute for Geophysics, Astrophysics, and Meteorology/Institute of Physics, University of Graz, Graz, Austria
[3]Zentralanstalt für Meteorologie und Geodynamik (ZAMG), Vienna, Austria

*Correspondence to:* Barbara Angerer
(barbara.angerer@uni-graz.at)

**Abstract.** The demand for high-quality atmospheric data records, applicable in climate studies, is undisputed. Using such records requires knowledge of the quality and the specific characteristics of all contained data sources. The latest version of the Wegener Center (WEGC) multi-satellite GPS radio occultation (RO) record, OPSv5.6, provides globally distributed upper-air satellite data of high quality, usable for climate and other high-accuracy applications. The GPS RO technique has

been deployed in several satellite missions since 2001. Consistency among data from these missions is essential to create a homogeneous long-term multi-satellite climate record. In order to enable a qualified usage of the WEGC OPSv5.6 dataset we performed a detailed analysis of satellite dependent quality aspects from 2001 to 2016. We present the impact of the OPSv5.6 quality control on the processed data and reveal time-dependent and satellite specific quality characteristics. Highest quality is found for MetOp (Meteorological Operational satellite) and GRACE (Gravity Recovery and Climate Experiment). Also data

from Formosat-3/COSMIC (Formosa Satellite mission-3/Constellation Observing System for Meteorology, Ionosphere, and Climate) are of high quality, however comparatively large day-to-day variations and satellite dependent irregularities need to be taken into account when using these data. We validate the consistency among the various satellite missions by calculating monthly mean temperature deviations from the multi-satellite mean, inducing also a correction for the different sampling characteristics. The results are highly consistent in the altitude range from 8 km to 25 km with mean temperature deviations

less than 0.1 K. At higher altitudes the OPSv5.6 RO temperature record is increasingly influenced by the characteristics of the bending angle initialization, with the amount of impact depending on the receiver quality.

## 1 Introduction

Detailed knowledge of the characteristics and the quality of data is essential for their qualified usage in atmospheric research. This also applies to the use of Global Positioning System (GPS) radio occultation (RO) data, which play a major role in the

characterization of the free atmosphere (Anthes, 2011). GPS RO is a limb sounding satellite technique, which has continuously provided atmospheric profiles since 2001. The technique uses the signal transmitted by Global Navigation Satellite System



(GNSS) satellites (in this work we only use RO data from GPS satellites) and received by low Earth orbit (LEO) satellites to probe the Earth's atmosphere.

The GPS signals are emitted at two radio frequencies in the L band (wavelengths of about 20 cm) and refracted on their way through the atmosphere. The LEO receiver measures the excess phase path due to the Earth's refractivity field which is proportional to density, in regions where humidity is negligible. Due to the relative motion of the satellites, the atmosphere is scanned vertically, either from top downwards for setting occultations (satellites move away from each other) or from bottom up for rising occultations (satellites move towards each other).

An RO event lasts about one to two minutes. Since the basic measurement of RO is the signal phase as function of time, external calibration is not needed and only short-term measurement stability is required over an RO event, which is ensured by the utilization of highly stable oscillators. The traceability to the international time standard (SI second) ensures long-term stability which is an essential prerequisite for climate applications (Leroy et al., 2006; Ohring et al., 2005).

The technique was first exploited in 1995 with the proof-of-concept mission Global Positioning System/Meteorology (GPS/Met). Highest data quality is obtained in the upper troposphere to lower stratosphere (UTLS) region (Scherllin-Pirscher et al., 2011b). Global coverage, long-term stability, high vertical resolution, and weather independence due to the GPS frequency in the microwave domain are further advantages (Kursinski et al., 1997). With these properties GPS RO has a significant impact on numerical weather prediction (e.g., Healy and Thépaut, 2006; Cucurull and Derber, 2008) and on our ability to monitor the atmospheric climate system (Anthes, 2011; Steiner et al., 2011).

For climate applications, data consistency and quality are essential for producing a homogeneous long-term multi-satellite record. Due to its self-calibrating nature, data from different RO missions and different sensor types can be combined to a consistent multi-satellite climate record, if the same processing system is used (Hajj et al., 2004; Schreiner et al., 2007; Foelsche et al., 2011).

Hajj et al. (2004) showed that data from the CHAMP (CHAllenging Minisatellite Payload) mission and SAC-C (Satélite de Aplicaciones Científicas-C) are remarkably consistent. Schreiner et al. (2010) compared co-located profiles from Formosat-3/COSMIC (Formosa Satellite mission-3/Constellation Observing System for Meteorology, Ionosphere, and Climate; F3C hereafter) satellites and confirmed that the root-mean-square (RMS) difference between 10 km and 20 km altitude is less than 0.2 % in refractivity. Foelsche et al. (2011) showed that refractivity and temperature climate records from multiple RO missions are consistent within 0.05 %, if the same processing scheme is applied.

Differences in the processing methods lead to structural uncertainties. This has been investigated in detail for RO data from CHAMP, processed by six different RO data centers (Ho et al., 2012; Steiner et al., 2013), finding high inter-center consistency, especially in the altitude range of about 8 km to 25 km altitude and within latitudes from 50° S and 50° N.

Besides the importance of consistent RO data, validating the quality of the individual satellite data is essential to identify atmospheric profiles with reduced quality and to assure the suitability of the dataset for climate applications.

In this paper we focus on the description of the Wegener Center (WEGC) RO record Occultation Processing System version 5.6 (OPSv5.6) in terms of data quality and multi-satellite consistency. An overview on the data used in the OPSv5.6 retrieval is given in Sect. 2. In Sect. 3 we focus on the retrieval methods and the quality control conducted during the retrieval. The data





quality of individual satellite missions is discussed in Sect. 4, and Sect. 5 describes the steps towards a combined multi-satellite record. Summary and main conclusions are given in Sect. 6.

## 2   RO data

WEGC OPSv5.6 uses amplitude and excess phase data as well as precise orbit information (position and velocity vectors for
both the GPS and the LEO satellite) from the University Corporation of Atmospheric Research/COSMIC Data Analysis and Archive Center (UCAR/CDAAC) as input data for the retrieval of atmospheric variables. Data from CHAMP, SAC-C, GRACE (Gravity Recovery and Climate Experiment), C/NOFS (Communications/Navigation Outage Forecasting System), MetOp (Meteorological Operational satellite) and F3C, have been processed. Depending on the availability at the UCAR/CDAAC data archive, reprocessed or post-processed data are used.

The German mission CHAMP, operated by the Helmholtz-Zentrum Potsdam/German Research Center for Geosciences (GFZ) and launched in 2000, was the first mission to provide a continuous multi-year RO record (May 2001 to October 2008) (Wickert et al., 2001). The mission was equipped with the GPS receiver BlackJack, also referred to as TRSR-2 (TurboRogue Space Receiver 2), produced by the Jet Propulsion Laboratory (JPL). The receiver was a new generation instrument of the GPS/Met receiver TRSR and was able to track around 250 setting RO events per day.

The US/Argentinian mission SAC-C (Hajj et al., 2004) was in orbit from 2000 to 2013 and had the same BlackJack receiver mounted as CHAMP, although it was named GPS occultation and passive reflection experiment (GOLPE). However, SAC-C was the first mission where open-loop (OL) tracking was implemented, which also enabled the aquisition of rising signals for the first time (Ao et al., 2009). The ability to track the GPS signal during rising occultations significantly increases the number of observations.

The two satellites of the joint US/German twin-satellite mission GRACE (Beyerle et al., 2005; Wickert et al., 2005) were launched in 2002 into the same polar orbit with the main focus of observing the Earth's time-variable gravity field. Both satellites (GRACE-A and GRACE-B) carry a modified version of the BlackJack receivers used on CHAMP and SAC-C. RO measurements are performed by GRACE-A since 2006, when the receiver was switched on permanently. GRACE-B only provides measurements for some shorter time periods (July 2014 – December 2014, June 2015 – October 2015, and April 2016
– September 2016), when swapping maneuvers took place, making GRACE-B the trailing satellite.

The major aim of the US C/NOFS (de la Beaujardière et al., 2004) mission was the monitoring of ionospheric scintillation, but also RO measurements were performed. The mission was operating between 2008 and 2015, however RO data at UCAR/CDAAC are only available for some months in 2010 and 2011. The orbit design of the C/NOFS mission was chosen such that mostly the tropical regions were covered (inclination of 13°). The GPS receiver utilized on C/NOFS, called CORISS
(C/NOFS Occultation Receiver for Ionospheric Sensing and Specification), is also of TRSR heritage.

The six identical spacecrafts of the Taiwanese/US F3C mission were launched in April 2006 and since June 2006 UCAR/CDAAC have continuously provided occultation measurements for F3C (Anthes et al., 2008). Each satellite is equipped with an IGOR



(Integrated GPS Occultation Receiver) receiver, which is also based on the design of the JPL BlackJack receiver and is capable of tracking both setting and rising occultations, yielding around 500 tracked RO events per day.

The MetOp series (Luntama et al., 2008), operated by the European Organisation for the Exploitation of Meteorological Satellites (EUMETSAT), consists of three satellites, with two satellites (MetOp-A and MetOp-B) currently in orbit (early 2017). MetOp-A has been providing RO data since the end of 2007 and MetOp-B since spring 2013. Both satellites are circulating in a sun-synchronous, 98°-inclined orbit and carry a GNSS receiver for Atmospheric Sounding (GRAS) receiver, which was jointly developed by Saab Ericsson Space (SES) of Sweden and Austrian Aerospace (AAE), now RUAG space. The four dual-frequency channels of the GRAS receiver allow two rising and two setting events to be tracked simultaneously, yielding a comparatively high number of around 700 observed RO events per day.

Table 1 lists the UCAR/CDAAC data used for the WEGC OPSv5.6 record, the UCAR/CDAAC data versions and available time periods as well as their launch date, mounted receiver and the differencing method applied to remove potential clock errors. The latest WEGC processed month is currently December 2016, however the OPSv5.6 record will be extended based on new UCAR/CDAAC data becoming available.

By mid-2017, only GRACE, both MetOp satellites and two out of six F3C satellites (F3C FM1 and F3C FM6) are still providing data. With the impending end of the F3C mission, new missions providing RO data are urgently needed. The succession mission of F3C, named Formosat-7/COSMIC-2 is scheduled to be launched in early 2018. The designated end of lifetime of the GRACE mission is in late 2017. However, the launch period for the GRACE-FO (GRACE Follow-On) mission will be between December 2017 and February 2018, with first data available approximately three months after the launch, leaving the data gap comparatively small. The launch of the third satellite of the MetOp series, MetOp-C, is currently planned for October 2018.

Furthermore, the Chinese FengYun-3 (FY-3) meteorological satellite series and commercial missions are expected to provide RO data to the international scientific and operational community soon. FY-3C was launched in 2013 and FY-3D is scheduled for launch in September 2017. Both missions carry the GNSS radio occultation sounder GNOS (Liao et al., 2016; Bai et al., 2017). Together with the planned exploitation of GNSS signals from the Russian GLONASS, the European Galileo system, and the Chinese BeiDou system this will enhance the number of RO observations in the future.

## 3 RO retrieval and quality control

In the following we describe the WEGC OPSv5.6 retrieval processing chain and the approach to assess the quality of the retrieved atmospheric parameters.

### 3.1 OPSv5.6 retrieval

OPSv5.6 includes a combined geometrics optics (GO)/wave optics (WO) retrieval of bending angle profiles. Input data are UCAR/CDAAC excess phase and amplitude profiles from occulted GNSS signals as well as precise orbit data of the GPS and LEO satellites. To reconstruct excess phase in the lower troposphere, navigation message information is also used from



UCAR/CDAAC. Detailed information on the OPSv5.6 retrieval chain was recently given by Schwärz et al. (2016), where we summarized the steps and related key aspects relevant for the quality analysis of this study.

Before entering the bending angle retrieval, quality checks, which comprise technical aspects and consistency of the input data, are performed (see Sect. 3.2 for more details). The bending angle is then calculated separately for the two GPS signal

frequencies L1 (= 1575.42 MHz) and L2 (= 1227.60 MHz) after calculating Doppler profiles from the excess phases (e.g., Melbourne et al., 1994; Kursinski et al., 1997). Subsequently the ionospheric influence on the excess phase measurement is removed by applying the ionospheric correction, where the L1 and L2 bending angle profiles are linearly combined (Vorob'ev and Krasil'nikova, 1994). This yields the ionosphere-corrected bending angle profile. The tropospheric bending angle is obtained from a WO retrieval following Gorbunov (2002) and Gorbunov and Lauritsen (2004). The transition from GO to WO

bending angle is performed between 7 km and 13 km.

To validate the quality of the retrieved bending angle in the upper atmosphere where the contribution from the neutral atmosphere is small, bending angle bias and noise are determined between 65 km and 80 km (Li et al., 2015). The bending angle bias is defined as the difference between the mean ionosphere-corrected RO bending angle and the mean Mass Spectrometer and Incoherent Scatter Radar (MSIS) bending angle in this height layer (Pirscher, 2010). The MSIS reference climatology is

used with fixed solar activity to avoid the influence of solar variations. The bending angle bias (with respect to MSIS) is usually slightly negative, with typical values of about $-0.1$ μrad (Danzer et al., 2013). Large systematic biases of the bending angle can indicate systematic errors in the satellites orbits, velocities, clock bias estimates, or incomplete removal of the ionospheric contribution to the measurement (Kursinski et al., 1997; Danzer et al., 2013).

Bending angle noise between 65 km and 80 km is defined as the standard deviation of the difference between the ionosphere-

corrected RO bending angle and the MSIS bending angle shifted by the bias (Pirscher, 2010; Li et al., 2015). Typically bending angle noise is smaller than 5 μrad. Larger bending angle noise can result from measurement noise due to poor GPS receiver quality onboard the LEO satellite, large residual ionospheric noise from ionospheric irregularities, or from the differencing method used to remove clock errors (e.g. Gorbunov, 2002; Gobiet et al., 2007). The latter is already performed on raw measurement data at UCAR/CDAAC.

Table 1 shows that single-differencing (SD) has been applied for most satellite data used in this study. SD involves another (second) GNSS satellite as reference link (Schreiner et al., 2010), which adds additional ionospheric noise. If ultra-stable oscillators are used onboard the LEO satellite (like on GRACE or MetOp), clock errors are so small that high-quality data can be obtained with zero-differencing (ZD), which avoids additional ionospheric noise (Beyerle et al., 2005); for more details and comparisons of both methods see, e.g., Schreiner et al. (2010) and Bai et al. (2017).

The next retrieval step relates the bending angle to the refractivity by using an Abel transform. Since this integral transform extends to infinity, initialization at high altitudes is needed. We use co-located ECMWF short-range forecasts, and MSIS above the uppermost ECMWF level, for the bending angle initialization. These co-located ECMWF profiles are extracted from the model by using the time layer closest to the mean event time and interpolated to the mean RO event location. Hence the OPSv5.6 RO data are not completely independent from ECMWF at high altitudes. The weight of the RO measurement relative

to the background information is determined based on the quality of the retrieved ionosphere-corrected RO bending angle. The





statistical optimization uses an inverse covariance weighting technique (Gobiet and Kirchengast, 2004; Gobiet et al., 2007) and is performed between 30 km and 120 km impact altitude.

The ratio of the retrieval error (influenced by both the observational and background error) and the error of the background determines the amount of background information contained in the statistically optimized profile. This ratio is denoted as retrieval-to-apriori error ratio (RAER), and the impact altitude where this ratio reaches 50 % is called zRAER50. Since the estimated observational error is a constant absolute value, the value of the zRAER50 is driven by its magnitude and the background error, a constant relative value (Gobiet et al., 2007). It indicates the transition altitude between background- and observational dominated height regions. In case of small observational errors the observation-dominated region will extend higher up into the stratosphere. For more details see Rieder and Kirchengast (2001).

Recently the observation-to-background weighting ratio (rOBW) has been introduced as a quantity more directly reflecting the fraction of observational information (Li et al., 2015; Schwarz et al., 2017). However, zRAER50 as used in OPSv5.6 is still a valuable indicator for the fractionation of information, in particular for inter-comparing amongst different missions.

After the statistical optimization and the retrieval of refractivity profiles, atmospheric variables are calculated. Neglecting the atmospheric wet term, dry density is calculated from atmospheric refractivity by applying the Smith-Weintraub formula (Smith and Weintraub, 1953). Dry pressure profiles are then retrieved using the hydrostatic equation and dry temperature profiles are subsequently obtained by the application of the ideal gas law (Kursinski et al., 1997).

The retrieval of the wet (physical) atmospheric variables is done in a simple version of the 1D-Var retrieval, where a priori knowledge of the state of the atmosphere is required. Co-located ECMWF short-range forecast profiles are again used as background data. A more detailed description of the OPSv5.6 retrieval can be found in Schwärz et al. (2016).

## 3.2 OPSv5.6 quality control

Quality assessment of the WEGC OPSv5.6 data is done in three major steps, as illustrated in Fig. 1. First, the quality of the UCAR/CDAAC input data is checked prior to the bending angle retrieval, to ensure that the retrieval can be performed. The input quality control (QC) rejects measurements if the accuracy of the time vector is not within 0.002 seconds. Furthermore the signal duration must be greater than 15 seconds and the straight-line tangent point of the occultation event has to be available within 20 km and 65 km impact altitude, otherwise the profile will be discarded.

The second step of the quality assessment is the internal QC where the quality of the retrieved ionosphere-corrected bending angle profile is examined. If the quality of the bending angle is not sufficient, i.e., the bending angle noise is greater than 22 μrad or the bending angle bias is greater than 10 μrad, the profile is discarded. Each profile that passes the internal QC gets marked with a bending angle quality flag (QF) according to its quality level and is processed further to atmospheric variables. Bending angle QF can be set to 0, 2, or 5 in the internal QC, where QF = 0 marks high bending angle quality. A detailed description on the meaning of the quality flags is given in Table 2.

Apart from detecting and eliminating profiles with insufficient bending angle quality, the bending angle QF provides also information on the weight of the RO measurement in the statistically optimized bending angle, being directly related to the observational error magnitude used in statistical optimization. For profiles with bending angle QF = 0, the observational error





is set to the value of the bending angle noise. Worse bending angle quality of profiles (QF = 2 or QF = 5), requires to use a larger amount of background information. The observational error is therefore set to a larger value (22 µrad), leading to more background information and less observational information in the retrieved variables.

External QC is the third step in the OPSv5.6 QC. It is conducted after finishing the retrieval of all atmospheric parameters. In
this step the plausibility of the retrieved atmospheric profiles of bending angle, refractivity, dry temperature, and temperature is examined by comparing them to co-located ECMWF analysis profiles. In this stage of the QC profiles are not discarded, but flagged with a QF that marks bad quality (QF = 1), if the deviations exceed a certain limit (see Table 2). In case of the bending angle, a QF has already been set in the internal QC.

If the external QC fails for the bending angle, the bending angle QF is updated and, depending on the result of the internal
QC, the new QF can be 10, 12, 15, 20, 22, or 25 (see Table 2 for details), where the one's digit indicates the internal QC and the ten's digit the external QC. As an example, QF = 25 means that the internal QF = 5 (negative bending angles below 50 km) and the external QF = 20 (retrieved bending angle profiles somewhere contain values outside of −1 mrad and 10 rad). In addition to the individual QFs of the retrieved atmospheric parameters, the so-called profile QF is defined. This QF is only 0 if all QFs are 0.

An OPSv5.6 output profile is hence always flagged with five quality flags: bending angle QF, refractivity QF, dry temperature QF, temperature QF and an overall profile QF. Only if the profile QF = 0, the profile is denoted as an OPSv5.6 high-quality profile and is recommended to be used for all general applications. However, depending on the user's needs, profiles with other QFs can also be of particular interest.

## 4   Quality aspects of the individual satellites

Knowledge of the differences in quality of the various satellite data is essential, especially if data from several missions shall be combined to a multi-satellite record. Figures 2 and 3 illustrate the quality of RO data from different missions as identified by OPSv5.6. Figure 4 illustrates the spatial distribution of RO event locations of all processed RO missions with their respective internal bending angle QF for one specific month as well as the latitudinal distribution of the internal bending angle QFs over the complete time range per satellite. Figure 5 comprises the information about the amount of high-quality profiles on a
monthly time scale for all satellites processed within the OPSv5.6 retrieval. In Table 3 an overview on the number of provided UCAR/CDAAC phase delays, the OPSv5.6 output profiles, and the high-quality profiles per satellite is given.

In the following, we discuss commonalities of as well as differences between the satellites before going into detail of satellite-specific features.

### 4.1   General features

The upper panels of Fig. 2 and Fig. 3 show the temporal evolution of the relative number of profiles passing various QC steps. The daily percentage is calculated relative to the total number of input files. In general, the input QC and the internal QC have the strongest impact on the number of high-quality profiles while the influence of the external QC is comparatively low. The



majority of all OPSv5.6 output profiles is flagged with QF = 0 as found in the lower panels of Fig. 2 and Fig. 3, however some irregularities and satellite specific characteristics can be detected. The percentage of high-quality profiles differs considerably among all satellites, which clearly points out the differences in individual satellite data quality. Besides that, data quality can also vary significantly over time.

5     The characteristics in the spatial event distribution, as shown in Fig. 4, reflect the different orbit designs of the receiving LEO satellites. The left panel shows the horizontal distribution of the internal QFs for one exemplary month (July 2008 and July 2011, respectively), indicating some seasonal effect at high latitudes. The right panel depicts the mean latitudinal distribution, revealing an equal distribution for both the Northern and Southern hemisphere for all satellites.

    Figure 5 reveals the vast increase in OPSv5.6 profiles in mid-2006, when RO data provision of the F3C mission was started.

10 A decline in the number of profiles can be observed in the last years, as some of the used RO missions already exceeded their designated lifetimes.

    The capability to track setting as well as rising signals is reflected in the number of provided measurements, as obtainable from Table 3. The lowest number can be found for CHAMP and GRACE, for which UCAR/CDAAC only provides setting measurements (although GRACE is capable of tracking also rising events). The F3C satellites and MetOp, which are capable

15 to track both kind of signals, provide twice (F3C) or even three times (MetOp) as much measurements. The number of output profiles varies between 57 % (C/NOFS) and a maximum of 94 % for GRACE. The majority of the missions obtains around 70 % of high-quality data.

## 4.2   CHAMP

% of all CHAMP data are rejected in the OPSv5.6 input QC and only 52 % of all UCAR/CDAAC input data yield a high-

quality output profile (Fig. 2a). Compared to the other satellites, this number is quite low. The daily percentage of profiles passing the different QC steps is constant over time, except at the beginning of the time series. A firmware update in March 2002 slightly increased the number of high-quality profiles. The difference between the total number of OPSv5.6 output profiles and the OPSv5.6 high-quality profiles can mainly be attributed to the high number of profiles flagged with QF = 2, primarily induced by negative bending angles between 50 km and 55 km. A semi-annual cycle is observable in the temporal evolution

of high-quality profiles because of the systematic rejection of very cold profiles (Schwarz, 2013).

    The CHAMP orbit has an inclination of 87°, which yields a quite homogeneous global distribution, with slightly less measurements in the tropics, see Fig. 4a. The number of profiles with QF = 2 and QF = 5 is greatest between 60° S and 90° S in July 2008 (Antarctic winter). Averaged over the complete time period (right panel of Fig. 4a) profiles with QF = 2 and QF = 5 are equally distributed above both poles. This can be understood as follows: Between 50 km and 55 km the absolute

atmospheric bending angle is small. Due to the comparatively high noise of CHAMP data it can happen that some values in the bending angle profile are negative. This occurs more likely in very cold regions, where the absolute bending angle values are particularly small. This effect is visible for all satellites, however it is strongest for CHAMP.



### 4.3 GRACE

The data quality of the GRACE mission is very good. There is almost no loss due to input QC and 80 % of all input data yield high-quality profiles (Fig. 2b). Furthermore data quality is constant over time, only a slight decline in high-quality profiles is visible which might be attributable to a degrading instrument performance after exceeding its planned lifetime. There is no

apparent change in data quality between the twin satellites, as both are equipped with the same receiver (time periods where RO measurements where taken by GRACE-B instead of GRACE-A are marked in Fig. 2b).

The number of provided UCAR/CDAAC phase delay files is significantly lower for GRACE than for other satellites (Table 3): Around 200 events per day, in contrast to, e.g. the average number of around 400 events per day for F3C FM5. Since the input QC rejects only very few of these events, measurements of lower data quality are presumably rejected already at a

previous processing step.

Due to its polar orbit (89° inclined) the occultation event locations are evenly distributed over the globe (Fig. 4b). The same spatial pattern of QF distribution (i.e., high number of QF $= 2$ and QF $= 5$ profiles at high latitudes due to the rejection of very cold profiles) can be observed for GRACE as for CHAMP, but in a reduced strength.

### 4.4 SAC-C

The number of high-quality profiles varies strongly in the beginning of the mission, but from 2006 onwards quality remains constant with time with an average number of 71 % high-quality profiles (Fig. 2c). From late 2002 on, the newly developed OL tracking mode was tested on SAC-C, to enable the tracking of signals in rising occultation. A stable version was then established in March 2006 (Ao et al., 2009). In the testing phase between 2003 and 2006, UCAR/CDAAC does not provide any measurement data.

Data prior to the OL testing phase have been processed with an older UCAR/CDAAC data version (Table 1) and because of the strongly varying data quality during this period we do not recommend to use SAC-C data before 2006. Again, the semi-annual cycle of the percentage of high-quality measurements (Fig. 2c) as well as the latitudinal dependency of the QFs (Fig. 4c) induced by the rejection of very cold profiles is visible, similar to CHAMP and GRACE.

### 4.5 C/NOFS

The C/NOFS mission was operating from 2008 to 2015, however UCAR/CDAAC only provides post-processed C/NOFS measurements from 2010 to 2011. More than 40 % of all C/NOFS input data are discarded in the input QC (Fig. 2d), because many events are too short and do not cover the altitude range from 20 km to 65 km. However, although there seems to be a problem in the vertical availability of C/NOFS measurements, the majority of the profiles that pass internal QC (48 % of all input data) are of high quality (see lower panel of Fig. 2d).

Because of the near-equatorial orbit of C/NOFS (inclination of 13°), RO measurements are only available at low latitudes up to 30° (Fig. 4d). Due to its focus on low latitudes, RO data from the C/NOFS mission can be valuable for studies concerning the tropical region, where the density of RO measurements is generally lower.





### 4.6 MetOp

Compared to the other satellite missions, the number of MetOp RO measurements is significantly higher. Around 600 events per day can be detected since MetOp is able to perform two rising and two setting occultation measurements simultaneously. MetOp-A shows a remarkably good and constant data quality, especially until mid-2013. From mid-2013 onwards the num-

ber of data passing the input and internal QC diminishes (from around 85 % to 75 %), yielding a total average of 80 % high-quality profiles. A tracking parameter update, which was performed in June 2013 on the MetOp-A GRAS receiver (C. Marquardt/EUMETSAT, personal communication, 2017), is reflected in the statistics of the QC, showing a decline in the number of profiles passing input QC as well as internal QC.

In contrast to the other satellites, the external QC has the largest impact for MetOp-A before mid-2013 (Fig. 2e). MetOp-B

also shows high data quality (Fig. 2f). A decline in the number of data passing input and internal QC occurs in April 2013, when the tracking parameter update has been applied for MetOp-B. Amongst other things, this update introduced a change in the tracking of the L2 signal: it is now measured from 15 km straight-line-tangent altitude (about 20 km impact altitude) upwards only. Since the OPSv5.6 uses the bending angles between 15 km to 20 km to extend the ionosphere correction in the troposphere this update implicates that this correction can not be applied. If the lowermost ray is above 20 km impact altitude

the profile is rejected by the QC completely.

The temporal evolution of the percentage of high-quality profiles of both MetOp satellites is constant with almost no outliers (Figs. 2e and 2f). Figure 4e illustrates the global coverage of MetOp-A occultations, with a slightly increased coverage in the mid-latitudes, which is attributed to the orbit inclination of 98.7°. Since MetOp-A and MetOp-B are operating in the same orbit, we only show results from MetOp-A in Fig. 4e as they are representative for both MetOp satellites.

### 4.7 Formosat-3/COSMIC

The six F3C satellites show similar characteristics in data quality throughout their active time periods, see Fig. 3. Distinct increases in input data quality are induced by firmware updates of the GPS receivers, e.g., in August 2006 and in January 2012.

The daily number of measurements varies strongly with time. There are also significantly stronger variations in the temporal evolution of high-quality profiles compared to the other satellites. These variations can mainly be attributed to the internal QC.

The semi-annual cycle of the percentage of high-quality profiles mainly stems from the rejection of bending angle profiles with a bending angle noise exceeding 22 µrad. All satellites also reveal a divergent behavior in the very beginning of the mission, between April and July 2006 (before the first firmware update took place), where the number of profiles flagged with QF = 2 is significantly higher than in the subsequent time period. Strong variations in data quality also appear in 2011, especially for F3C FM4. The time period is much shorter for F3C FM3, as it has been out of operation already since August 2010.

Global coverage is also achieved for the F3C mission (see Fig. 4f for FM1, which is representative for the other F3C satellites), with slightly less measurements near the poles and in the tropics. For a better coverage of the tropics, the Formosat-7/COSMIC-2 mission, which is the successor mission of F3C, will have a constellation of 6 satellites at 24° inclined orbits. In addition, six satellites are planned in a near-polar orbit with 72° inclination (Schreiner, 2016).





## 5  Towards a combined multi-satellite record

The unique properties of RO, including high accuracy and long-term stability, can be exploited to create a consistent multi-satellite climate record. To ensure a high-quality and consistent multi-satellite OPSv5.6 RO record, we first inspect the respective bending angle characteristics to identify unusual behavior that would lead to inconsistencies in the combined dataset. We
then consider differences in the sampling characteristics of the various missions and analyze deviations from the multi-satellite mean in retrieved temperature time series.

### 5.1  Bending angle consistency

To validate the quality and consistency of OPSv5.6 data, we analyze bending angle bias, standard deviation of the bending angle noise (for brevity just termed noise hereafter), and the altitude where the retrieval-to-apriori error ratio equals 50 %
(zRAER50). All these parameters are defined within the OPSv5.6 retrieval chain (see Sect. 3.1 for how they are estimated) and characterize the quality of the retrieved bending angle. Therefore, they are suitable quantities for validating data consistency, as already shown by Pirscher (2010), Foelsche et al. (2011), and Schwärz et al. (2016).

In Fig. 6a we show the temporal evolution of the daily median bias for all satellites processed within the OPSv5.6 retrieval. For climate applications it is important that the bending angle bias is similar for all satellites and close to zero, which is true for
all satellites. The slightly negative values are mainly attributed to residual ionospheric effects (Danzer et al., 2013). No distinct inhomogeneities are visible over time and the mean values vary between $-0.09\,\mu\mathrm{rad}$ for SAC-C and $-0.20\,\mu\mathrm{rad}$ for C/NOFS. These slightly different mean values might result from data being available from different time periods.

C/NOFS data, for example, are only available in 2010 and 2011, when solar activity was high, and only cover the lower latitudes, where Total Electron Content (TEC) is comparatively high. High solar activity causes a higher level of ionization
in the Earth's upper atmosphere, which again affects the quality of RO measurements (i.e., a larger ionospheric residual and therefore a larger bending angle bias). This has been empirically shown by Schreiner et al. (2011) and Danzer et al. (2013) and underpinned by end-to-end simulations including atmospheric and ionospheric models by Liu et al. (2015).

The temporal evolution of the daily median bending angle noise is depicted in Fig. 6b. The bending angle noise, which mainly reflects the quality of the GPS receiver and the residual clock errors, is largest for CHAMP with $4.00\,\mu\mathrm{rad}$. The
BlackJack receiver mounted on CHAMP is a receiver of the first generation (Sect. 2), which explains the comparatively high noise for CHAMP. Bending angle noise is significantly lower ($2.54\,\mu\mathrm{rad}$) for GRACE, which also utilizes a BlackJack receiver, however an already advanced version. In addition, zero-differencing can be applied to account for the receiver clock error due to ultra-stable oscillators used in the GRACE mission, which also leads to less noisy data (Beyerle et al., 2005). We assume that zero-differencing has been applied for GRACE first in the UCAR/CDAAC 2014.2760 data version, as the noise value becomes
smaller in the time period after April 2014.

The smallest bending angle noise is found for the two MetOp satellites, with $0.90\,\mu\mathrm{rad}$ for MetOp-A and $0.95\,\mu\mathrm{rad}$ for MetOp-B, which reflects the excellent quality of the MetOp/GRAS receiver. SAC-C and all six F3C satellites reveal large temporal variability of noise in the early times of their mission lifetimes. Due to these large fluctuations, that match the





results shown in Sect. 4, data from these time periods (2001-08 to 2002-11 for SAC-C and 2006-04 to 2006-07 for F3C) are not included in OPSv5.6 multi-satellite climatologies; in the subsequent time periods the noise amounts to about 2.5 μrad, including single-differencing, for F3C and around 3 μrad for SAC-C.

Figure 7 shows that zRAER50 for MetOp is at a considerably higher impact altitude than for all other satellites. For F3C a small fraction of profiles reveals zRAER50 values at higher impact altitude. The assumed change to zero-differencing for GRACE in the newer data version (2014.2760) is reflected in the increased zRAER50 value in April 2014. The zRAER50 parameter (Sect. 3.1) is strongly influenced by the quality of the satellite's receiver. Because of the high quality of the MetOp/GRAS receiver, the influence of the ECMWF background field on the MetOp observations is far smaller than for any other satellite at the same altitude level, confirming early theoretic/simulation studies such as by Rieder and Kirchengast (2001) and Steiner and Kirchengast (2005).

Figure 8 shows the impact of the statistical optimization on the monthly mean bending angle profiles of F3C FM2 and MetOp-A at 45 km impact altitude, where the RAER is already at around 50 % for F3C FM2, but still significantly lower for MetOp. Differences between the non-optimized bending angle and the statistically optimized bending angle for both satellites fluctuate around zero (violet dots), however F3C FM2 shows larger variations than MetOp. The influence of background information on statistically optimized F3C FM2 bending angles can best be seen during significant changes in the ECMWF system (e.g. mid-2012 and mid-2013). Since ECMWF data are used as background information in the statistical optimization (see Sect. 3.1), the difference between non-optimized and statistically optimized bending angle shows some (small) jumps during these time periods.

In comparison, such changes in ECMWF are less visible in the MetOp time series. From this we can conclude that due to the high quality of the Metop/GRAS receiver, the high-altitude initialization (and with that, ECMWF model system changes) has less impact on retrieved MetOp profiles compared to other missions. This has to be kept in mind when generating a combined multi-satellite record from all RO missions.

## 5.2 Monthly mean multi-satellite climatologies

When combining data from different satellite missions to a global multi-satellite record, not only the quality and consistency of the retrieved atmospheric profiles, but also the differences in spatial and temporal sampling have to be taken into account. The error due to discrete sampling (sampling error) can be estimated from the difference between the averaged co-located ECMWF analysis profiles and the averaged full ECMWF field (Foelsche et al., 2008; Pirscher et al., 2010). In order to account for the sampling error, it is subtracted from the climatology (e.g., Foelsche et al., 2011; Steiner et al., 2011), leaving a small residual sampling error (for detailed information see Scherllin-Pirscher et al. (2011a)).

The impact of the sampling error correction is clearly visible in Fig. 9. These monthly mean global mean dry temperature differences are calculated for each satellite relative to the multi-satellite mean between 8 km and 25 km height following Foelsche et al. (2011) and Steiner et al. (2011). If no sampling error correction is applied (9 top), deviations mainly vary between 0.5 K with large outliers occurring in mid-2011 (out of plot scale). These outliers stem from large deviations of C/NOFS which only provides data for the tropics. Not considering these special spatial sampling characteristics leads to a bias





in the global mean. With the sampling error correction applied (9 bottom), deviations are below 0.1 K for all satellites in the 8 km to 25 km height range where RO is of highest quality.

At higher altitudes, between 25 km and 35 km, MetOp-A reveals a distinctively different behavior from the other satellites before mid-2013, but a consistent one after that (Fig. 10). This change in the characteristics of MetOp dry temperature devi-
ations coincides precisely with an ECMWF model system change (cycle 38r2), where, among other changes, the number of vertical levels was increased in the model.

Several other ECMWF model system changes can be identified in the deviations from each satellite to the ECMWF analysis field in this altitude range, see Fig. 10. We find that the improvements of the model are reflected in decreasing deviations of the RO data from ECMWF after 2013. This is specifically evident for MetOp, which is generally less ECMWF-affected in this
altitude range than the other satellites, and which shows the largest decrease in deviation from ECMWF. With the increase in quality of the ECMWF model system, the analysis approaches the high-quality atmospheric information provided by MetOp, resulting in better quality of the other, more ECMWF-affected, satellites.

At lower altitudes up to 25 km the impact of the high-altitude initialization on the RO temperature is small, consequently different RO missions can be readily combined to a multi-satellite dataset (see Fig. 9 and Steiner et al. (2011); Foelsche et al.
(2011)). Above 25 km, the RO temperature record is increasingly influenced by the characteristics of the initialization, with the amount of impact depending on the receiver quality. This effect is most pronounced for temperature, since the hydrostatic integration in the retrieval step from refractivity (density) to pressure leads to a downward propagation of high altitude inital-ization errors, propagating further into temperature. High consistency in refractivity can be achieved up to higher altitudes of about 30 km (Foelsche et al., 2011).

**6   Summary and conclusions**

In this study, we performed a quality analysis of the individual satellite datasets comprising the latest version of the Wegener Center multi-satellite GPS RO record WEGC OPSv5.6. We described the QC procedure applied in the OPSv5.6 retrieval and included a detailed analysis of the impact of the various QC steps on each individual satellite dataset for the missions CHAMP, GRACE, SAC-C, C/NOFS, MetOp, and Formosat-3/COSMIC. A rigorous QC is key for establishing a combined multi-satellite
record with known properties and shall facilitate proper application of the WEGC OPSv5.6 record.

From this analysis, we conclude that our bending angle quality control passes for CHAMP a less-than-average number of events as high-quality profiles compared to the other satellites. The six F3C satellites have a comparable, high bending angle quality with quite large day-to-day variations and some satellite-specific irregularities. GRACE and especially the two MetOp satellites show the highest quality, however the MetOp time series reflects relevant changes in the receiver software.
The improvement of the receiver quality with time from the older BlackJack receivers to the modified BlackJack receiver on GRACE, and the GRAS receiver on MetOp, is evident in our analysis.

As the signal-to-noise ratio in the RO observations gets worse with increasing altitude the observations are merged with background information at high altitudes in the bending angle retrieval. The impact of the background field is strongly depen-





dent on the quality of the receiver and increases with decreasing receiver quality. For establishing a homogeneous long-term RO climate record it is thus essential to track the quality and influence of the background field to understand the height-dependent characteristics.

In the OPSv5.6 retrieval ECMWF short-term forecasts are used as background. At higher altitude levels the influence of

the background field increases, and with that certain ECMWF model changes are reflected in the OPSv5.6 retrieval results. In the RO retrieval chain this impact propagates further downwards for each retrieved parameter. We conclude that the OPSv5.6 multi-satellite refractivity/temperature record is only marginally influenced by the ECMWF model up to about 30 km/25 km, and higher up for MetOp due to its excellent receiver quality.

The WEGC OPSv5.6 record provides a valuable observational record for atmospheric analysis and climate monitoring.

Based on the knowledge from our careful quality control we find high-quality temperature data from different satellites highly consistent between 8 km to 25 km, with deviations from the multi-satellite mean of less than 0.1 K. Above, the quality of individual satellite records depends on receiver quality and the amount of background information that comes in. The MetOp record provides high-quality observational information to higher altitudes which is reflected in a larger divergence from the multi-satellite mean between 25 km to 35 km. Improvements in the quality of the background field in 2013 led to much smaller

deviations of less than 0.2 K. These findings have to be taken into account when using the WEGC OPSv5.6 RO record above about 25 km to 35 km for climate trend applications.

This work aided to enhance the maturity of the RO record with respect to knowledge of data quality and its description (Bates and Privette, 2012), and helps to meet the stringent requirements as defined by the Global Climate Observing System (GCOS, 2010a, b, 2011) for the generation of a climate data record of essential climate variables.

*Data availability.* Dataset is currently available on request from the authors and will be made publicly available soon via an online resource.

*Competing interests.* The authors declare that they have no conflict of interest.

*Acknowledgements.* We are grateful to UCAR/CDAAC (Boulder, CO, USA) for the provision of its RO excess phase and orbit data (available at http://cdaac-www.cosmic.ucar.edu/) and ECMWF (Reading, UK) for providing access to analysis and forecast data (available at http://www.ecmwf.int/en/forecasts/datasets). This work was funded by the Austrian Science Fund (FWF) under grant P27724-NBL (VER-

TICLIM) as well as by the FFG-ALR projects OPSCLIMTRACE (ASAP-9 844395) and OPSCLIMVALUE (ASAP-10 848013).



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





**Table 1.** Excess phase and orbit input data to the WEGC OPSv5.6 radio occultation data processing. Regarding the differencing methods SD denotes single-differencing and ZD denotes zero-differencing.

| Mission | Launch | GPS receiver | UCAR processor version | Time period | Differencing method |
|---|---|---|---|---|---|
| **CHAMP** | 2000 | BlackJack | 2014.0140 | 2001-05-18 to 2008-10-05 | SD |
| **GRACE** | 2002 | BlackJack | 2010.2640 | 2007-02-28 to 2014-03-30 | SD |
| | | | 2014.2760 | 2014-03-31 to 2016-09-30 | ZD |
| **SAC-C** | 2000 | BlackJack | 2005.3090 | 2001-08-13 to 2002-10-14 | SD |
| | | | 2005.1720 | 2002-11-03 to 2002-11-15 | |
| | | | 2010.2640 | 2006-03-09 to 2011-08-03 | |
| **C/NOFS** | 2008 | CORISS | 2010.2640 | 2010-03-01 to 2011-12-31 | SD |
| **MetOp-A** | 2006 | GRAS | 2016.0120 | 2007-09-30 to 2016-10-31 | ZD |
| **MetOp-B** | 2012 | GRAS | 2016.0120 | 2013-02-01 to 2016-10-31 | ZD |
| **F3C FM1** | 2006 | IGOR | 2013.3520 | 2006-04-23 to 2014-04-30 | SD |
| | | | 2014.2050 | 2014-05-30 to 2014-06-29 | |
| | | | 2014.2860 | 2014-05-01 to 2016-11-29 | |
| **F3C FM2** | 2006 | IGOR | 2013.3520 | 2006-05-01 to 2014-04-30 | SD |
| | | | 2014.2860 | 2014-05-01 to 2016-09-23 | |
| **F3C FM3** | 2006 | IGOR | 2013.3520 | 2006-04-24 to 2010-07-05 | SD |
| **F3C FM4** | 2006 | IGOR | 2013.3520 | 2006-04-22 to 2014-04-30 | SD |
| | | | 2014.2050 | 2014-06-01 to 2014-06-29 | |
| | | | 2014.2860 | 2014-05-01 to 2015-07-07 | |
| **F3C FM5** | 2006 | IGOR | 2013.3520 | 2006-04-28 to 2014-04-30 | SD |
| | | | 2014.2050 | 2014-06-01 to 2014-06-27 | |
| | | | 2014.2860 | 2014-05-01 to 2016-04-16 | |
| **F3C FM6** | 2006 | IGOR | 2013.3520 | 2006-04-22 to 2014-04-30 | SD |
| | | | 2014.2050 | 2014-06-01 to 2014-06-29 | |
| | | | 2014.2860 | 2014-04-30 to 2016-11-29 | |





**Table 2.** Quality flags (QFs) defined within the WEGC OPSv5.6 data processing.

| Variable | Flag | Type | Meaning |
|---|---|---|---|
| **Bending angle** | **QF = 0** | internal QC | All checks passed. Bending angle retrieval results are of high quality. |
| | **QF = 2** | internal QC | Bending angle noise could not be calculated from bending angle profile. A large observational error (22 µrad) is used in the retrieval. |
| | **QF = 5** | internal QC | Negative bending angles below 50 km. A large observational error is used in the retrieval. Only non-optimized bending angle profiles should be used. |
| | **QF = 10** | external QC | If internal QF = 0 but the relative difference between the retrieved non-optimized bending angle profile and the co-located ECMWF analysis bending angle profile is greater than 20 % somewhere between 10 km and 35 km. |
| | **QF = 12** | external QC | If internal QF = 2 and relative difference to co-located ECMWF exceeds limit (see QF = 10). |
| | **QF = 15** | external QC | If internal QF = 5 and relative difference to co-located ECMWF exceeds limit (see QF = 10). |
| | **QF = 20** | external QC | Retrieved bending angle profiles somewhere contain values outside of −1 mrad and 10 rad. |
| | **QF = 22** | external QC | If internal QF = 2 and bending angle profiles contain values outside defined range (see QF = 20). |
| | **QF = 25** | external QC | If internal QF = 5 and bending angle profiles contain values outside defined range (see QF = 20). |
| **Refractivity** | **QF = 0, 1** | external QC | If the relative difference between retrieved RO refractivity profile and the co-located ECMWF analysis refractivity profile is greater than 10 % somewhere between 5 km and 35 km, QF is set to 1. Else, QF is 0. |
| **Dry temperature** | **QF = 0, 1** | external QC | If the difference between the retrieved RO dry temperature profile and the co-located ECMWF analysis dry temperature profile is greater than 20 K somewhere between 8 km and 25 km, QF is set to 1. Else, QF is 0. |
| **Temperature** | **QF = 0, 1** | external QC | If the difference between the retrieved RO physical temperature profile and the co-located ECMWF analysis physical temperature profile is greater than 20 K somewhere between 8 km and 25 km, QF is set to 1. Else, QF is 0. |
| **Profile** | **QF = 0, 1** | external QC / internal QC | Profile QF defines the high-quality profiles. If the QFs of each checked variable is 0, profile QF is set to 0, else to 1. |



**Table 3.** Overview on daily average number of available UCAR/CDAAC excess phase files (one file per RO event) and the corresponding percentages of retrieved OPSv5.6 output profiles and high-quality profiles.

| Mission | Average no. of UCAR/CDAAC excess phase files / day | OPSv5.6 output profiles [%] (all QFs) | OPSv5.6 high-quality profiles [%] (QF=0) |
|---------|------|------|------|
| CHAMP   | 206  | 78%  | 52%  |
| GRACE   | 197  | 94%  | 80%  |
| SAC-C   | 245  | 83%  | 71%  |
| C/NOFS  | 250  | 57%  | 48%  |
| MetOp-A | 650  | 87%  | 80%  |
| MetOp-B | 643  | 78%  | 72%  |
| F3C FM1 | 467  | 79%  | 72%  |
| F3C FM2 | 353  | 80%  | 74%  |
| F3C FM3 | 407  | 77%  | 70%  |
| F3C FM4 | 429  | 78%  | 72%  |
| F3C FM5 | 409  | 82%  | 75%  |
| F3C FM6 | 393  | 71%  | 62%  |





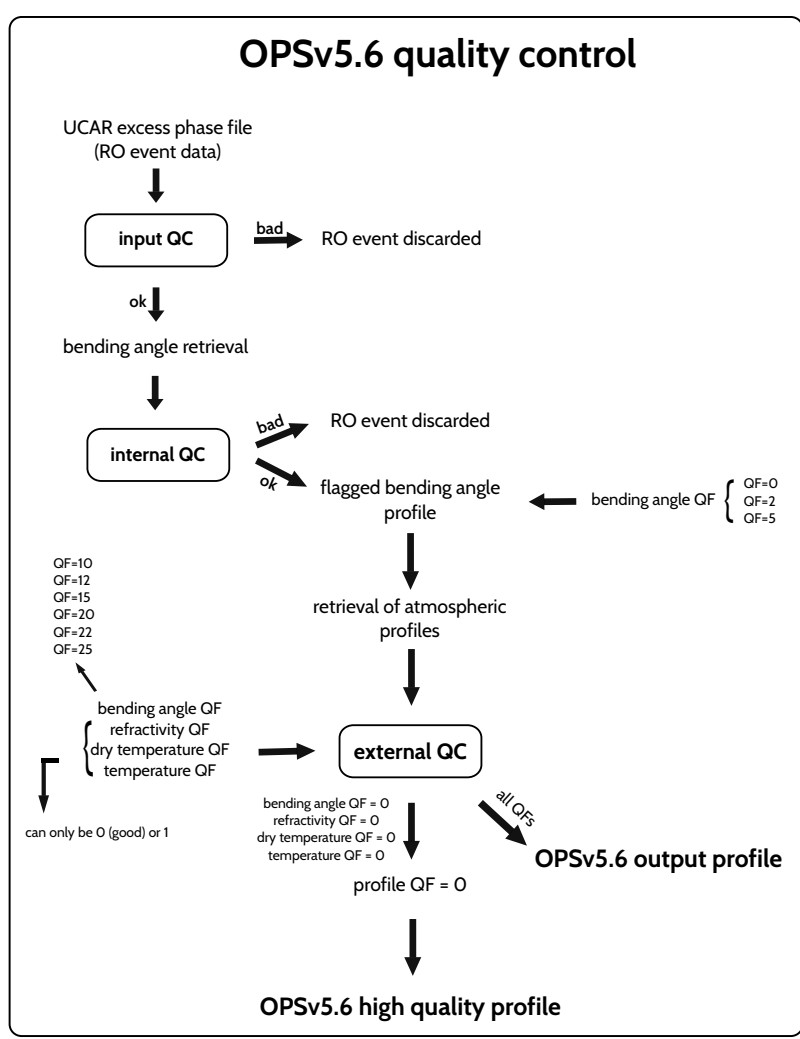

**Figure 1.** Schematic representation of the quality control approach of the WEGC OPSv5.6 retrieval.





**Figure 2.** Temporal evolution of daily percentage of profiles (relative to total number of input data) passing the various QC steps for (a) CHAMP, (b) GRACE, (c) SAC-C, (d) C/NOFS, (e) MetOp-A and (f) MetOp-B (upper panels). Upper sub-panels of (a) to (f) show the number of profiles passing the input QC (blue dots), the number of bending angle profiles passing internal QC, which equals the number of output profiles (green dots), the number of profiles that are within the defined limits in the external QC (bending angle QF is 0, 2, or 5 and all atmospheric QF are 0) (red dots) and the daily percentage of high-quality OPSv5.6 profiles (orange dots). Daily percentage of flagged profiles, relative to the number of OPSv5.6 output profiles, is shown separately for each bending angle QF in the lower sub-panels.





**Figure 3.** Temporal evolution of daily percentage of profiles (relative to total number of input data) passing the various QC steps for the six F3C satellites, (a) FM1, (b) FM2, (c) FM3, (d) FM4, (e) FM5 and (f) FM6. Upper sub-panels of (a) to (f) show the number of profiles passing the input QC (blue dots), the number of bending angle profiles passing internal QC, which equals the number of output profiles (green dots), the number of profiles that are within the defined limits in the external QC (bending angle QF is 0, 2, or 5 and all atmospheric QF are 0) (red dots) and the daily percentage of high-quality OPSv5.6 profiles (orange dots). Daily percentage of flagged profiles, relative to the number of OPSv5.6 output profiles, is shown separately for each bending angle QF in the lower sub-panels.





**Figure 4.** Spatial distribution of the RO events with bending angle QF=0, 2, or 5 for the satellites (a) CHAMP, (b) GRACE, (c) SAC-C, (d) C/NOFS, (e) MetOp-A and (f) F3C FM1, showing the geographical coverage for one exemplary month (left sub-panels) and the latitudinal distribution over the complete available time range per satellite (right sub-panels).



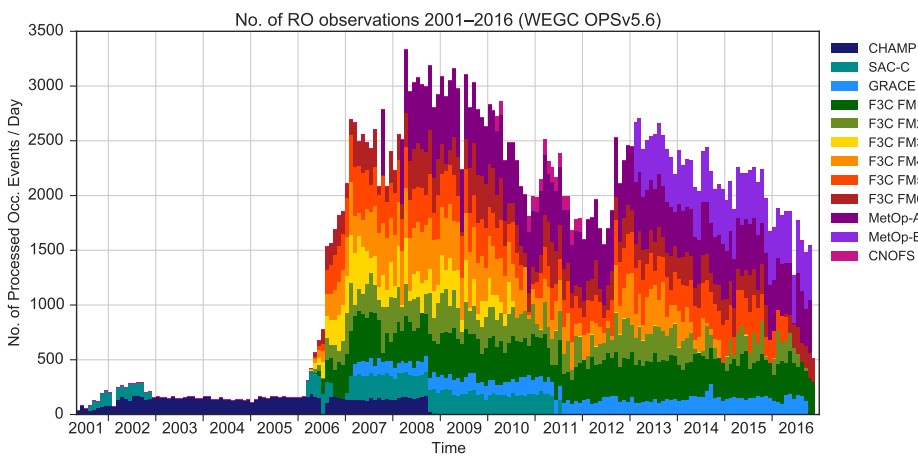

**Figure 5.** Daily number of high-quality OPSv5.6 profiles for different satellites (different colors) as a function of time from 2001 to 2016.





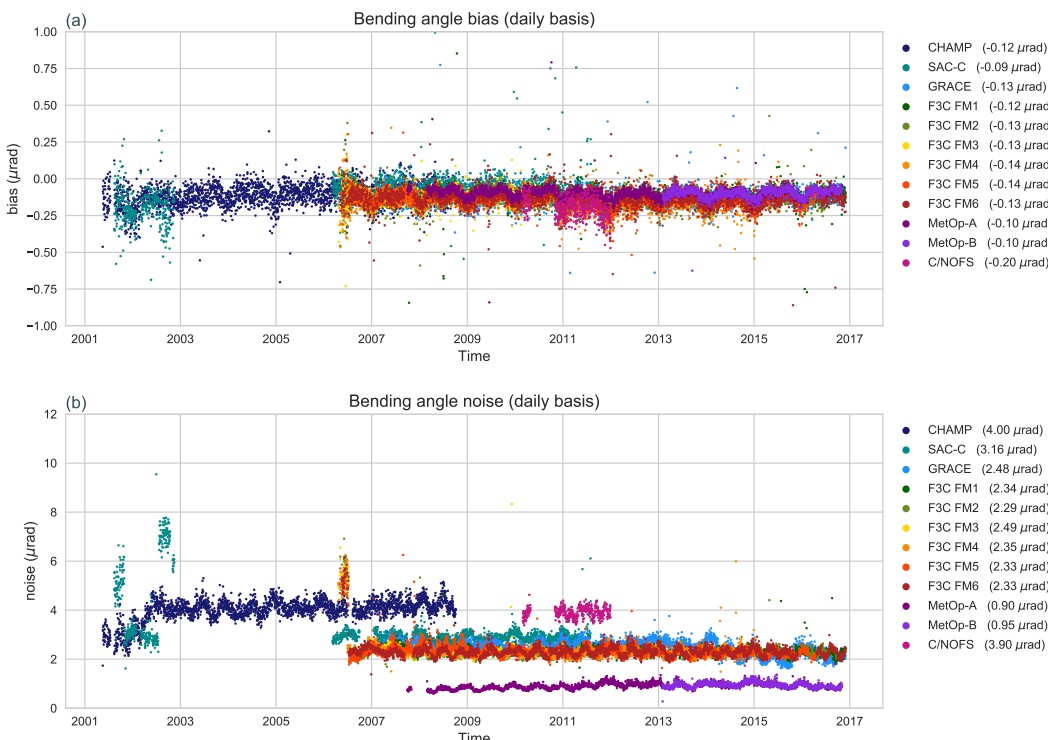

**Figure 6.** Temporal evolution of daily median bending angle bias (a) and bending angle noise standard deviation (b) for all RO missions used in OPSv5.6. Total mean of bending angle bias and noise is shown as value in parentheses in the respective legend. Only high-quality profiles are used in these statistics.





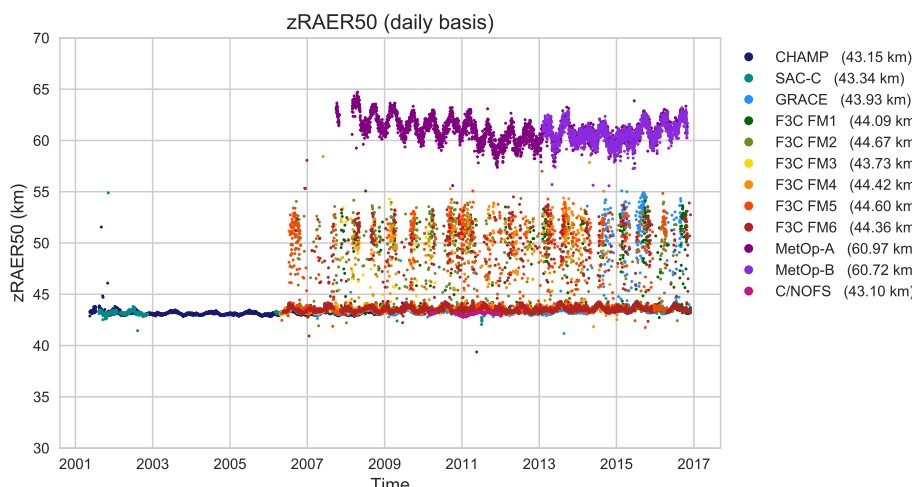

**Figure 7.** Temporal evolution of daily median zRAER50 for all RO missions processed in OPSv5.6. The mean of zRAER50 is shown as value in parantheses in the legend. Only high-quality profiles are used in these statistics.



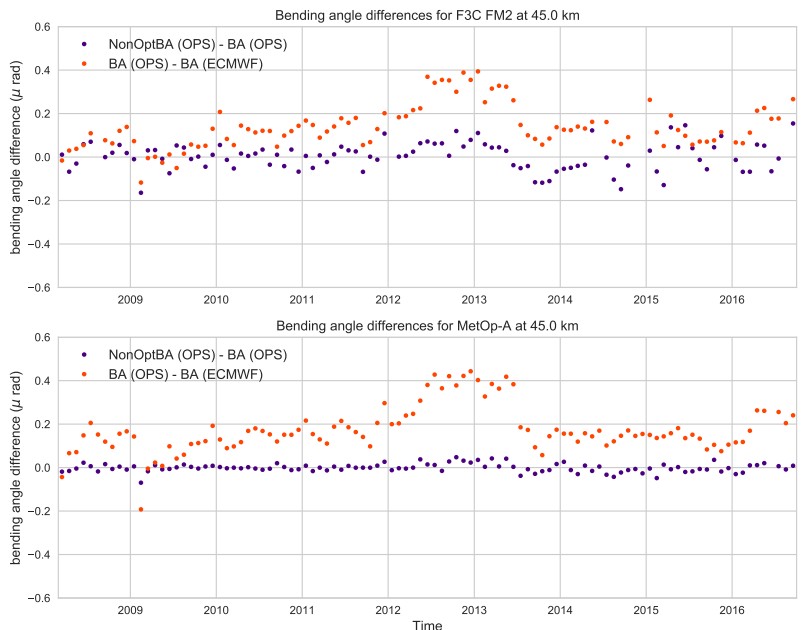

**Figure 8.** Monthly averaged differences between non-optimized (NonOptBA) and statistically optimized OPSv5.6 bending angle (BA) (violet dots) as well as between statistically optimized OPSv5.6 bending angle and co-located ECMWF bending angle (orange dots), shown for F3C FM2 (top) and for MetOp-A (bottom) at an impact altitude of 45 km.





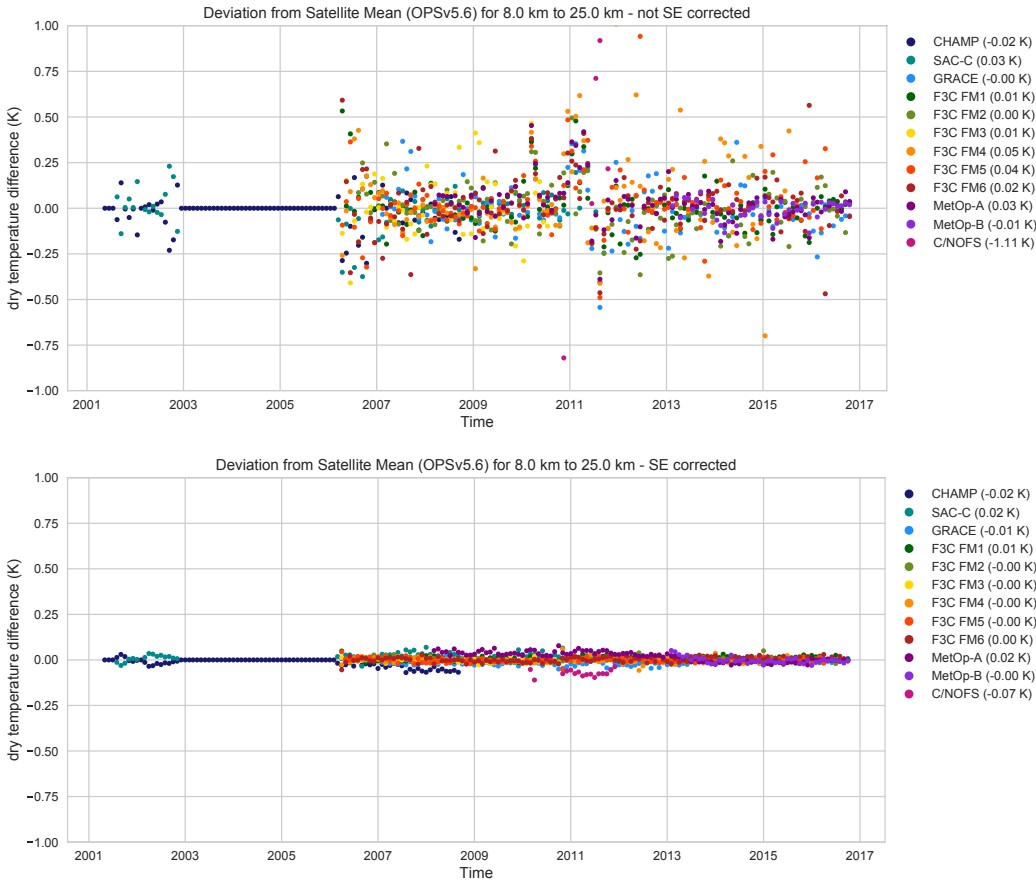

**Figure 9.** Deviations of individual satellites from the multi-satellite mean for monthly mean dry temperature in altitude layer 8 km – 25 km, without sampling error (SE) correction (top) and with SE correction applied (bottom).





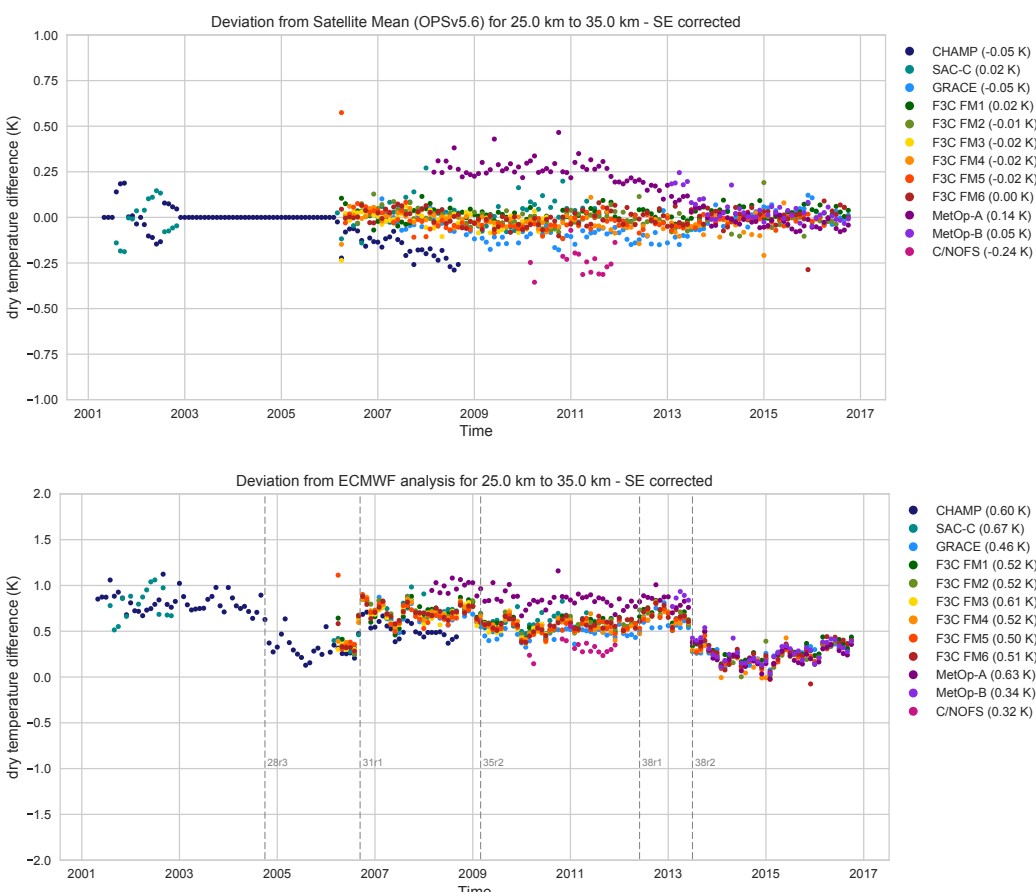

**Figure 10.** Deviations of individual satellites from the multi-satellite mean (top) and from the ECMWF analysis (bottom), for monthly mean dry temperature in the altitude layer 25 km – 35 km. Points in time (vertical dashed line) mark important ECMWF model system changes (bottom).