# Peer review of "Quality Aspects of the Wegener Center Multi-Satellite GPS Radio Occultation Record OPSv5.6"

_Atmospheric Measurement Techniques, 2017_

## Referee Comment (RC1) · Anonymous Referee #1 · 1 Sep 2017

**General comments**

The paper "Quality Aspects of the Wegener Center Multi-Satellite GPS Radio Occultation Record OPSv5.6" by Angerer et al. describes Wegener Center's RO processing system, with a focus on the quality control steps included in the processing chain, and on quality aspects of the resulting data records. Two factors in particular are discussed: time dependence and satellite/mission dependence. The paper leads up to a discussion of bending angle quality as a function of time and satellite, and a short discussion of the impact of sampling-error correction on dry temperature climatologies.

The paper is primarily of interest for those working on RO processing or in related

fields. However, the concluding discussions, in Section 5 and some of the Figures, should be of interest also to a wider audience. Many of the results should be useful to anyone working with, or planning to work with, long-term RO climate data records.

For those themselves working on RO processing there are many interesting details discussed: the QC procedures is an issue that turns out to be important, and the overview of the RO satellite missions, instruments, and differencing methods is quite useful. With this overview as background, the time series showing QC statistics and bending angle quality give a good overall picture of the level of consistency amongst the RO data sets.

The paper is well suited for publication in AMT.

*Specific comments*

(i) Much of the quality control and quality monitoring is based on the bias and standard deviation (noise) of the bending angle profile in the height interval 65-80 km. What vertical resolution does the raw ionospheric-corrected bending angle profiles have? Is it just corresponding to the excess-phase sampling frequency? Note that the noise may be affected by this resolution, and also by filtering of the excess-phase time series. Any comments on this issue?

(ii) One way to minimize time dependencies when generating long-term climate data records is to take a priori information from reanalyses rather than from operational NWP models. What are the considerations here?

(iii) The sampling errors are estimated as the difference between the sub-sampled and the full ECMWF field. It seems lika a good practice to subtract these errors from the observed climatology, and the results clearly show that it is efficient in removing a large fraction of sampling-related artefacts. But is there any risk that one accidentally add something to the climatology that shouldn't be added? Suppose you have a bin,

in which you do your averaging, and in your model there is an overall gradient across the bin whereas in the real atmosphere there is no gradient. With a non-uniform sampling, your estimated sampling error would include a component that shouldn't be there. The same type of reasoning could be made for, e.g., a diurnal cycle that is not perfectly described by the model. Any comments on these risks, and the suitability of the chosen approach?

---

## Referee Comment (RC2) · Anonymous Referee #2 · 29 Sep 2017

General comment: This is a carefully drafted manuscript, giving plenty of details regarding the processing of GPS RO data. Only minor revisions are recommended.

MInor comments:

page 4 line 30: It is left unclear to the reader why a combined wave optics/geometric optics retrieval is used. It should be briefly mentioned why a "pure" wave optics approach, while more general, is suboptimal.

page 8 line 14: It is mentioned that only setting events from GRACE are available from UCAR. Are rising events from GRACE available elswhere? page 10 line 31-33: This sentence about future developments does not really fit in this section that describes

already taken measurements. It would fit better in the introduction where the future missions are described, or in the conclusions.

section 5.1,5.2: It is described in some detail how the ECMWF background influences the bending angle profiles and that there is clear influence on the raw profiles at high altitudes, probably less so in the optimized profiles. Jumps in the background as seen in Figs 8,10 have an imprint on the raw bending angle profiles and are not guaranteed to be eliminated in the optimized profiles. In this respect it is somewhat puzzling why operational ECMWF background information is used although likely much more homogeneous background information from reanalyses, (ERA-Interim or JRA55) is available as well. This way one could reduce the potential for inhomogeneities in the RO retrievals at very high altitudes. It should either be explained why operational ECMWF background data have been preferred over reanalyses or, even better, sensitivity experiments should be performed using background data from reanalyses.

Fig.7: Why is the zRAER so constant for CHAMP, and why isn't it lower for CHAMP than for other platforms, given its significantly higher noise level?

Fig. 8 and following figures: Axis labels are very small

Fig. 9, 10 top: Strictly speaking a deviation from the multi satellite mean is not defined if only one data source is present and even with 2 platforms (SAC-C and CHAMP in 2001/2) it cannot be reliably estimated. I would start this plot in 2006. In its present form it is misleading since it suggests better quality before 2006 compared to after 2006, which is not the case.

---

## Author Comment (AC1) · 6 Oct 2017

We thank the reviewer for the affirmative feedback and the valuable comments. Please find our response below:

(comment 1) "Much of the quality control and quality monitoring is based on the bias and standard deviation (noise) of the bending angle profile in the height interval 65-80 km. What vertical resolution does the raw ionospheric-corrected bending angle profiles have? Is it just corresponding to the excess-phase sampling frequency? Note that the noise may be affected by this resolution, and also by filtering of the excess-phase time

(response 1) In our processing, the excess phase profiles are smoothed using a regularization filter. The vertical resolution of the raw ionosphere-corrected bending angle profile is primarily determined by the filter width used there (to a minor degree also by the additional L2 signal filtering during the ionospheric correction). This yields a vertical resolution of about 2 km for the ionosphere-corrected bending angle in the mesosphere where we extract these standard deviation (noise) diagnostics, leading to a certain noise level for bending angle profiles that is comparatively smaller than for the excess phases. The key point for making the bending angle noise a useful diagnostic is, however, that we use the filter settings in a fixed way for all multi-satellite processings and over the full time period. Therefore it is always the same excess phase-to-bending angle processing that is applied, independent of the specific RO mission (CHAMP, COSMIC, MetOp, etc.). The magnitude of the bending angle noise for the different RO missions is hence a good diagnostic to help judge the mission performance and the degree of influence of the upper boundary initialization.

We added the following sentence in the manuscript:

**page 5 line 4**: "Before entering the bending angle retrieval, the excess phase is filtered using a regularization filtering method, with identical filter settings for all RO missions."

(comment 2) "One way to minimize time dependencies when generating long-term climate data records is to take a priori information from reanalyses rather than from operational NWP models. What are the considerations here?"

(response 2) Thank you for pointing this out. We are aware that using a priori information from reanalyses for our bending angle initialization would possibly be preferable for climate applications. However, most of the reanalyses assimilate RO
and are therefore not independent of RO. Furthermore, even reanalyses records are not free of influences due to observing system changes, or other biases. This is why we use ECMWF forecasts (24 h and 30 h), which are largely independent of RO. We added the following sentence to clarify this:

**page 5 line 32**: "Using ECMWF forecasts (24 h and 30 h), instead of, e.g., ECMWF analyses, prevents the direct impact of assimilated RO data on the high altitude initialization. The forecast range of at least a day is sufficient to make the a priori information decorrelated from the analyses information."

(comment 3) "The sampling errors are estimated as the difference between the sub-sampled and the full ECMWF field. It seems lika a good practice to subtract these errors from the observed climatology, and the results clearly show that it is efficient in removing a large fraction of sampling-related artefacts. But is there any risk that one accidentally add something to the climatology that should not be added? Suppose you have a bin, in which you do your averaging, and in your model there is an overall gradient across the bin whereas in the real atmosphere there is no gradient. With a non-uniform sampling, your estimated sampling error would include a component that should not be there. The same type of reasoning could be made for, e.g., a diurnal cycle that is not perfectly described by the model. Any comments on these risks, and the suitability of the chosen approach?"

(response 3) We agree that there is a risk of introducing a bias from the chosen reference field to the sampling error corrected climatology. However, as mentioned in the manuscript (page 12 line 28), this residual sampling error is estimated to be small. Scherllin-Pirscher et al. (2011a) conducted an error analysis on climatologies and concluded for the residual sampling error to be of the same order of magnitude as the statistical error. Nevertheless, we intend to investigate a possible bias due to the SE
correction in the future, e.g., by using different reference fields.

---

## Author Comment (AC2) · 6 Oct 2017

We thank the reviewer for the positive comments and the constructive questions. Please find our response below:

**(comment 1) page 4 line 30**: "It is left unclear to the reader why a combined wave optics/geometric optics retrieval is used. It should be briefly mentioned why a "pure" wave optics approach, while more general, is suboptimal."

[Figure]

**(response 1)** The use of the wave optics method (implemented in our processing system in form of the CT2 method) at higher altitudes than just the troposphere would indeed improve the vertical resolution also at stratospheric altitudes, but would consequently also lead to a higher noise level. Since we do not focus to provide data for process studies, such as high vertical resolution wave studies which would benefit from 100 m-scale resolution, but rather focus on climate-related applications, a vertical resolution of 0.5 km to 1 km is considered sufficient. This is just achievable also by the geometric optics method (as its "half-Fresnel scale" resolution limit). Therefore we chose this combined GO/WO retrieval (with transition near or somewhat below the tropopause) as suitable for our purposes.

We changed/added the following sentences in the manuscript:

**page 4 line 30** "OPSv5.6 includes a combined geometrics optics (GO)/wave optics (WO) retrieval of bending angle profiles, with transition from GO to WO near or somewhat below the tropopause. A combined GO/WO bending angle retrieval approach, yielding a vertical stratospheric resolution of 0.5 km to 1 km and low noise level, is suitable for the targeted OPSv5.6 data usage purposes."

**(comment 2) page 8 line 14**: "It is mentioned that only setting events from GRACE are available from UCAR. Are rising events from GRACE available elsewhere?"

**(response 2)** We are not aware that rising events of GRACE are available somewhere. We changed the comment on the rising occultations in order to avoid misunderstandings.

**page 8 line 14**: "The lowest number can be found for CHAMP and GRACE, for which only setting measurements are available."

[Figure]

**(comment 3) page 10 line 31-33**: "This sentence about future developments does not really fit in this section that describes already taken measurements. It would fit better in the introduction where the future missions are described, or in the conclusion."

**(response 3)** We agree with this comment and have moved the sentence of **page 10 line 31-33** to section 2 (section concerning current and future RO missions), **page 4 line 16**.

**(comment 4)** "section 5.1, 5.2: It is described in some detail how the ECMWF background influences the bending angle profiles and that there is clear influence on the raw profiles at high altitudes, probably less so in the optimized profiles. Jumps in the background as seen in Figs 8,10 have an imprint on the raw bending angle profiles and are not guaranteed to be eliminated in the optimized profiles. In this respect it is somewhat puzzling why operational ECMWF background information is used although likely much more homogeneous background information from reanalyses, (ERA-Interim or JRA55) is available as well. This way one could reduce the potential for inhomogeneities in the RO retrievals at very high altitudes. It should either be explained why operational ECMWF background data have been preferred over reanalyses or, even better, sensitivity experiments should be performed using background data from reanalyses."

**(response 4)** Thank you for that comment. This has also been an issue in the comments of Referee No. 1. For convenience, we repeat here the answer to Referee No. 1:

We are aware that using a priori information from reanalyses for our bending angle initialization would possibly be preferable for climate applications. However, most of

the reanalyses assimilate RO and are therefore not independent of RO. Furthermore, even reanalyses records are not free of influences due to observing system changes, or other biases. This is why we use ECMWF forecasts (24 h and 30 h), which are largely independent of RO. We added the following sentence to clarify this:

**page 5 line 32**: "Using ECMWF forecasts (24 h and 30 h), instead of, e.g., ECMWF analyses, prevents the direct impact of assimilated RO data on the high altitude initialization. The forecast range of at least a day is sufficient to make the a priori information decorrelated from the analyses information."

**(comment 5)** "**Fig.7**: Why is the zRAER so constant for CHAMP, and why isn't it lower for CHAMP than for other platforms, given its significantly higher noise level?"

**(response 5)** Thank you for pointing this out. The zRAER value is driven by the magnitude of the observational error, which is set according to the quality of the bending angle. In the case of QF=0 profiles, the observational error can either be set to the value of the bending angle noise, or set to a constant value (4.5 $\mu$rad). For CHAMP, the observational error is set to this value for nearly all profiles and also for COSMIC this is still the case for many profiles (but less than for CHAMP) (see Fig. 1). This is reflected in the constant median shown in Fig. 7 of the manuscript. To better illustrate the mean differences of zRAER50 between the different missions, we have now replaced the median in Fig. 7 of the manuscript with the daily mean of zRAER50 (see also Fig. 2).

We changed the following sentence in the manuscript:

**page 7 line 1**: "For profiles with bending angle QF=0, the observational error is set to the value of the bending angle noise, or a constant value (4.5 $\mu$rad, empirically determined) for profiles with degrading bending angle quality above 65 km."

**(comment 6)** "Fig. 8 and following figures: Axis labels are very small"

**(response 6)** We increased the font size of the axis labels.

**(comment 7) Fig. 9, 10 top**: "Strictly speaking a deviation from the multi satellite mean is not defined if only one data source is present and even with 2 platforms (SAC-C and CHAMP in 2001/2) it cannot be reliably estimated. I would start this plot in 2006. In its present form it is misleading since it suggests better quality before 2006 compared to after 2006, which is not the case."

**(response 7)** Thank you for this comment. Even after 2006, the meaning of the satellite mean changes over time because not all missions provide data for the whole time period. For the early time period (2001/2), the term satellite mean is a bit misleading, but we think that the information about the difference between CHAMP and SAC-C is still of value. We would prefer to keep the whole timeline, and we have added the following sentence to the figure caption to point to this issue:

**page 30 figure caption Fig. 9** "The satellite mean is calculated from all missions available for the respective month (note that before May 2006 only CHAMP and SAC-C delivered data)."

**Bending angle ObsError (daily basis)**

- CHAMP  (4.47 $\mu$rad)
- SAC-C  (4.46 $\mu$rad)
- GRACE  (4.25 $\mu$rad)
- F3C FM1  (4.24 $\mu$rad)
- F3C FM2  (4.02 $\mu$rad)
- F3C FM3  (4.36 $\mu$rad)
- F3C FM4  (4.11 $\mu$rad)
- F3C FM5  (4.04 $\mu$rad)
- F3C FM6  (4.12 $\mu$rad)
- MetOp-A  (0.42 $\mu$rad)
- MetOp-B  (0.44 $\mu$rad)
- C/NOFS  (4.47 $\mu$rad)

**Fig. 1.**

**Bending angle zRAER50 (daily basis)**

Fig. 2.